# FEDEQUILIBRIA: TOWARDS FAIR AND ROBUST FEDERATED LEARNING UNDER DOMAIN SKEW

## ABSTRACT

Federated Learning (FL) has been widely emerged as a promising paradigm for decentralized machine learning. However, its practical effectiveness is hindered by domain skew, a prevalent form of data heterogeneity where clients hold statistically different data distributions. Existing studies have revealed that this failure is rooted in two fundamental problems: (1) update conflicts, arising when clients' learning objectives diverge, and (2) model aggregation bias, where conventional aggregation schemes neglect domain diversity, systematically favoring certain clients over others. To tackle these intertwined challenges, we propose FedEquilibria, a novel and fair federated aggregation framework. The core idea of FedEquilibria is to employ a server-side hybrid weighting mechanism consisting of two synergistic steps. First, it formulates aggregation as a multi-objective optimization problem, uniquely leveraging the Fisher Information Matrix (FIM) as a proxy for each client's empirical objective. By computing conflict-aware weights on the server-side, FedEquilibria identifies a Pareto-optimal consensus of parameters that are structurally important across different domains. Second, to counteract aggregation bias, FedEquilibria calculates drift-aware weights based on the Euclidean norm of client updates, explicitly quantifying the fitting gap for each client and adaptively increasing the influence of underrepresented domains. Comprehensive experiments on benchmark datasets with pronounced domain shifts demonstrate that FedEquilibria significantly surpasses existing state-of-the-art methods. It achieves not only higher average accuracy but also substantially enhances fairness by improving the performance of the worst-performing clients, offering a principled solution for robust and equitable models in real-world FL systems.

## 1 INTRODUCTION

Federated Learning (FL) offers a paradigm for training powerful machine learning models collaboratively across decentralized clients while preserving data privacy (Kairouz et al., 2021; Yang et al., 2019). The canonical algorithm, FedAvg (McMahan et al., 2017), aggregates local models on a central server to produce a global model, which is then broadcast back for subsequent rounds of training. However, the efficacy of FL is significantly challenged by data heterogeneity (Li et al., 2020; Kairouz et al., 2021), particularly under the condition of domain skew (Li et al., 2021b; Tan et al., 2022). This scenario, where clients' data are drawn from distinct feature distributions, often leads to inconsistent convergence and undermines the global model's generalization capabilities.

A critical consequence of domain skew is the degradation of performance fairness (Li et al., 2019; Mohri et al., 2019; Pan et al., 2024). Standard FL methods, by optimizing for a single global objective, tend to favor clients from easier or dominant domains, inadvertently neglecting those with more challenging or minority data distributions. This bias not only leads to a substantial performance disparity across clients but also discourages participation from underperforming parties, ultimately limiting the collective knowledge of the federation and contradicting the core tenets of collaborative learning. We attribute this fairness-performance trade-off to two fundamental issues:

(1) Update Conflict and Parameter Salience Disparity: In heterogeneous FL settings, clients frequently produce updates in conflicting directions. Moreover, naive local updating treats all parameters equally, even though certain parameters are more salient for specific domains (LeCun et al., 1989; Kirkpatrick et al., 2017). Under the domain skew scenario in FL, critical updates (*e.g.*, closely

related to their local tasks) from an underperforming client may be nullified or overshadowed by less significant updates from other clients. Such conflicts at the parameter level result in a global model that inadequately captures information from all participating domains, thereby reinforcing performance unfairness.

(2) Biased Aggregation and Client Drift: Conventional aggregation schemes, such as weighting by sample size, inherently introduce bias by overemphasizing data-rich clients or disproportionately amplifying the influence of data-scarce clients. This issue is due to static weighting failing to account for the diverse learning dynamics across domains. Consequently, some clients may converge faster than others, leading to a phenomenon known as client drift, where local models diverge significantly from the global empirical objective. Such client drift not only reduces the effectiveness of aggregation but also impedes the convergence of clients with more complex or heterogeneous data, thereby exacerbating performance disparities.

To address these challenges, we introduce FedEquilibria, a novel federated learning framework designed to promote fairness via a two-pronged strategy that reconciles update conflicts and corrects aggregation bias. FedEquilibria leverages the Fisher Information Matrix (FIM) to construct a parameter sensitivity profile for each client, framing the aggregation process as a multi-objective optimization problem within the FIM space. This formulation enables the server to compute aggregation weights that achieve a Pareto-optimal consensus on parameter importance, thereby harmonizing clients' learning objectives at a structural level. Subsequently, FedEquilibria employs a drift-aware weighting mechanism to correct aggregation bias and reduce the impact of client drift. By leveraging the Euclidean distance between each client's update and the global model as a proxy for the domain-specific fitting gap, FedEquilibria adaptively amplifies the contributions of underrepresented clients. Benefiting from these two mechanisms, FedEquilibria produces global updates in a stable and equitable manner, making the global model converge quickly and reach better performance.

In summary, FedEquilibria provides a principled solution to the fairness problem in FL under domain skew. It first identifies what matters (salient parameters via FIM) and then intelligently decides how to combine them (fair aggregation via MOO and distance-based drift control). Our main contributions are as follows:

- We introduce the use of the Fisher Information Matrix in a federated context to identify client-specific salient parameters, enabling an aggregation process that mitigates update conflicts and enhances fairness.
- We formulate the aggregation problem as a multi-objective optimization task, deriving client weights that seek a fair, consensus-based update direction, thereby preventing the global model from being biased towards any specific domains.
- We incorporate a drift-aware mechanism that amplifies the influence of under-represented clients by weighting their updates based on their L2-norm distance from the global model, directly counteracting model aggregation bias.
- We conduct extensive experiments on standard domain-skew benchmarks, such as Digits and Office-Caltech (Gong et al. (2012)), demonstrating that FedEquilibria significantly improves both average performance and fairness metrics compared to state-of-the-art methods.

## 2 RELATED WORKS

**Federated Learning under Data Heterogeneity.** Federated Learning (FL) fundamentally grapples with the challenge of non-IID data, where client data distributions vary significantly. A primary line of research addresses this by modifying the local training process. FedProx (Li et al., 2020) introduced a proximal term to the client's local objective, which restricts local updates from drifting too far from the global model, thereby improving convergence on heterogeneous data. Other works, such as SCAFFOLD (Karimireddy et al., 2020), tackle client drift by estimating and correcting for the update direction disparities between clients and the server. When heterogeneity manifests as domain skew, specialized methods have been proposed. FedBN (Li et al., 2021b), for instance, maintains client-specific Batch Normalization layers while sharing other parameters, effectively allowing each client to learn domain-specific feature statistics. Other approaches focus on enhancing alignment at the feature level. MOON (Li et al., 2021a) utilizes contrastive learning to pull local and

global model representations closer, while FedProto (Tan et al., 2022) learns and aggregates abstract class prototypes instead of the full model parameters to achieve a more robust consensus. Further advancements have introduced more sophisticated mechanisms to handle domain-skewed data. For instance, FedLsa (Fu et al., 2025) employs learnable semantic anchors and hyperspherical contrast at the representation level, while Pa3Fed (Huang & Liu, 2025) proposes a period-aware adaptive aggregation strategy, dynamically modifying client weights based on their performance changes during training.

**Fairness in Federated Learning.** Beyond mere performance on heterogeneous data, ensuring fairness—equitable performance across all clients—is a critical challenge. A significant body of work has focused on designing aggregation schemes that explicitly promote fairness. Agnostic Federated Learning (AFL) (Mohri et al., 2019) formulates the problem through the lens of robust optimization, learning a single global model that performs well for any possible mixture of clients, thereby optimizing for worst-case client performance. Another influential approach, q-Fair Federated Learning (q-FedAvg) (Li et al., 2019), modifies the FedAvg objective to give higher weight to clients with larger losses, thus prioritizing clients on which the model performs poorly. More recent works view FL through a multi-objective optimization lens, where each client represents a distinct objective. Methods like FedFV (Wang et al., 2021) and FedLF (Pan et al., 2024) mitigate gradient conflicts among clients to find a common descent direction that improves performance for all participants. Tackling the problem more directly under domain skew, FedHEAL (Chen & Vikalo, 2024) was recently proposed to alleviate parameter update conflicts and model aggregation bias, explicitly aiming to improve performance fairness by harmonizing important parameters during aggregation.

## 3 BACKGROUND

### 3.1 FEDERATED LEARNING

Federated Learning (FL) is a distributed machine learning paradigm that enables multiple clients to collaboratively train a global model without sharing their raw data (Kairouz et al., 2021). Formally, the goal of FL is to solve the following optimization problem: $\min_\theta F(\theta) := \sum_{k=1}^N p_k F_k(\theta)$, where $\theta$ represents the parameters of the global model, $N$ is the total number of clients, $p_k$ is the aggregation weight for client $k$ (typically proportional to its dataset size, such that $\sum_{k=1}^N p_k = 1$), and $F_k(\theta)$ is the local objective function for client $k$ computed over its private dataset $D_k$. The standard algorithm for solving this problem is FedAvg. In each communication round $r$, a central server first broadcasts the current global model $\theta_r$ to a subset of clients. Each selected client $k$ then performs multiple steps of local training to obtain an updated local model $\theta_k^{r+1}$. These local models are subsequently sent back to the server, which aggregates them to produce the next global model: $\theta_{r+1} = \sum_{k=1}^N p_k \theta_k^{r+1}$. While effective in ideal settings, the performance of this framework is highly sensitive to the underlying data distributions across clients, a challenge broadly known as data heterogeneity (Li et al., 2020).

### 3.2 DOMAIN SKEW

A particularly challenging form of data heterogeneity in FL is domain skew (Li et al., 2021b; Chen et al., 2024). This scenario arises when clients share the same set of labels but possess data with different underlying feature distributions. Formally, for any two clients $k$ and $j$, their marginal label distributions are identical, $P_k(y) = P_j(y)$, but their conditional feature distributions are different, $P_k(x|y) \neq P_j(x|y)$. A digit recognition task can involve clients with visually distinct domains such as MNIST (LeCun et al. (2002)), SVHN (Netzer et al. (2011)),USPS (Hull (2002)) and SYN (Ganin & Lempitsky (2015)), all of which share the same label space (digits 0-9) but exhibit vast differences in style, color, and background. This skew leads to two primary issues that degrade FL performance. First, it causes update conflicts, where the gradients computed on different domains point in divergent directions, making the simple averaging in FedAvg inefficient or even counterproductive (Wang et al., 2021; Karimireddy et al., 2020). Second, it leads to performance unfairness, as the global model tends to converge towards a solution that favors clients with easier or more dominant domains, while neglecting others (Mohri et al., 2019).

### 3.3 Fairness in Federated Learning

In the context of FL, our work focuses on performance fairness: the goal of training a single global model that exhibits uniformly high performance across all participating clients, ensuring that no single client is disproportionately disadvantaged by the collaborative training process (Li et al., 2019). Performance fairness is typically quantified using metrics that capture outcome disparity. A common metric is the standard deviation of accuracies across clients ($\text{Std}(\{Acc_k\})$), where a lower value indicates a fairer model. Another critical metric is the worst-client accuracy ($\min_k Acc_k$), as improving performance on the most challenging domains is a direct indicator of enhanced fairness (Mohri et al., 2019). Under domain skew, standard FL algorithms often exhibit poor fairness, as performance can vary dramatically from one domain to another, motivating the need for aggregation mechanisms that are explicitly fairness-aware (Pan et al., 2024).

## 4 Method

To tackle fairness under domain skew, we propose FedEquilibria, a federated learning framework that adaptively balances client contributions. The core idea of FedEquilibria lies in its server-side aggregation strategy, which intelligently weights client contributions by synergistically considering both the conflict of their learning objectives and the magnitude of their local updates. As illustrated in Figure 1, in each round, clients perform standard local training and compute two key pieces of information: their model updates and a profile of their parameter sensitivities via the Fisher Information Matrix. The server then uses this information to compute a hybrid aggregation weight for each client, ensuring a fair and stable update to the global model.

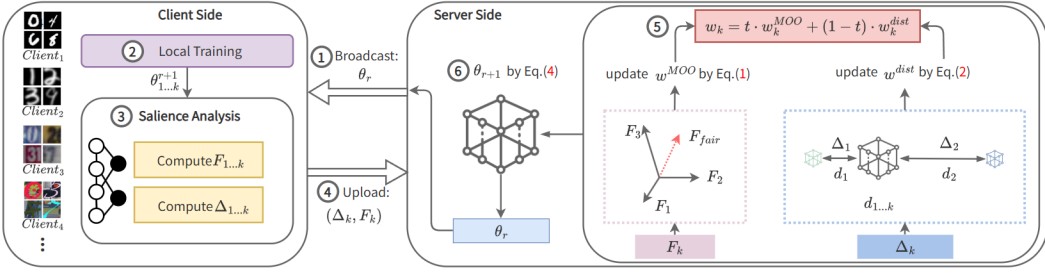

Figure 1: The overall architecture of FedEquilibria. Clients upload their model updates ($\Delta_k$) and Fisher Information Matrix profiles ($\mathbf{F}_k$). The server performs a two-stage aggregation: first, it uses the FIM profiles for Multi-Objective Optimization to achieve conflict reconciliation. Second, it uses the update norms for distance-based weighting to perform bias correction. The resulting hybrid weights are used to produce a fair global model.

### 4.1 Client-Side Computation

In each communication round $r$, every participating client $k \in S_r$ receives the current global model $\boldsymbol{\theta}_r$. After performing $E$ local training epochs on its private dataset $D_k$, the client computes its updated local model $\boldsymbol{\theta}_k^{r+1}$. From this, two components are prepared for upload to the server.

**Model Update.** The first component is the standard model update vector $\Delta_k$, which represents the full change in parameters after local training: $\Delta_k = \boldsymbol{\theta}_r - \boldsymbol{\theta}_k^{r+1}$.

**Parameter Sensitivity via FIM.** The second component is a measure of each parameter's sensitivity, which is crucial for the server's fairness-aware aggregation. One principled way to identify important parameters is to use the Fisher Information Matrix (FIM), as has been done in various contexts from network pruning to continual learning Kirkpatrick et al. (2017); Golatkar et al. (2020); LeCun et al. (1989). The sensitivity of the model's loss with respect to a parameter $\theta_i$ can be quantified by the second-order derivative of the loss function. This sensitivity can be interpreted as the importance of that parameter for the given task Maltoni & Lomonaco (2019). While directly computing the second derivative (the Hessian matrix) is often computationally prohibitive, a key insight is that the diagonal of the FIM is equivalent to the expectation of the second derivative of the nega-

tive log-likelihood (the loss) Pawitan (2001). Critically, the FIM can be efficiently computed using only first-order derivatives (gradients) Kay (1993); Aich (2021).

Following this well-established approach, we use the diagonal of the FIM, denoted as $\mathbf{F}_k$, as the *parameter sensitivity profile* for client $k$. It is computed over the client's local data $D_k$ using the locally updated model $\boldsymbol{\theta}_k^{\tau+1}$. The Fisher information for each parameter $\theta_i$ is given by: $\mathbf{F}_{k,ii} = \mathbb{E}_{(x,y)\sim D_k}\left[\left(\frac{\partial\mathcal{L}(y,f(x;\theta_k^{\tau+1}))}{\partial\theta_i}\right)^2\right]$, where $\mathcal{L}$ is the loss function. This vector $\mathbf{F}_k$ effectively captures which parameters are most critical for the client's specific data domain. The client then uploads the pair $(\Delta_k, \mathbf{F}_k)$ to the server.

## 4.2 SERVER-SIDE HYBRID AGGREGATION

The server's primary task is to compute a fair aggregation weight $w_k$ for each client. We introduce a hybrid weighting scheme that linearly combines two complementary signals: objective conflict and client drift.

**Conflict Reconciliation via Multi-Objective Optimization.** To mitigate the update conflicts that arise from domain skew, we reformulate aggregation as a Multi-Objective Optimization Problem (MOP), achieving consensus among competing objectives in federated learning (Hu et al., 2022b; Pan et al., 2024). However, prior works typically apply MOP directly to client gradients. We argue that this approach can be unstable, as gradients are often noisy and reflect only the instantaneous state of a single mini-batch.

Instead, our approach seeks a more fundamental consensus by operating in the space of parameter importance. We use the clients' FIM vectors $\mathbf{F}_k$ as stable, structural proxies for their domain-specific learning objectives. The goal is to find a set of aggregation weights $\{w_k^{MOO}\}$ that minimizes the norm of the aggregated FIM vectors. Geometrically, this corresponds to finding a point of minimal conflict energy in the parameter importance space, representing a Pareto-optimal consensus. We formulate this as a Quadratic Program (QP), a standard approach for solving the Multiple Gradient Descent Algorithm problem (Désidéri, 2012):

$$\{w_k^{MOO}\} = \arg\min_{\mathbf{w}}\left\|\sum_{k\in S_r} w_k\cdot\text{vec}(\mathbf{F}_k)\right\|_2^2, \tag{1}$$

subject to $\sum_{k\in S_r} w_k = 1$ and $w_k \geq 0$ for all $k$. Here, $\text{vec}(\cdot)$ denotes the vectorization operator. This QP problem can be efficiently solved using standard convex optimization solvers.

**Bias Correction via Update Norm.** While the MOO-based weighting effectively addresses conflicts among client objectives, it does not inherently solve the problem of Model Aggregation Bias Chen et al. (2024). Conventional FL aggregation schemes, whether based on data volume or even sophisticated conflict resolution, often neglect the underlying domain diversity of the clients. This neglect can lead to a biased global model that converges towards domains with easier convergence paths or greater representation, while marginalizing others. As argued in recent works, a larger change in a client's local model can signify a greater fitting gap between its local data and the current global model, indicating that its domain is being overlooked and possesses more potential for performance improvement.

Motivated by this, we introduce a drift-aware weighting component to explicitly account for domain diversity and counteract aggregation bias. We use the magnitude of a client's local update as a proxy for this fitting gap. The distance $d_k$, representing the degree of client drift, is defined as the L2 norm of the full model update: $d_k = \|\Delta_k\|_2$. A larger norm $d_k$ suggests that client $k$'s domain is more distinct from the current global consensus, and thus requires a larger adjustment to fit its data. By assigning weights proportionally to this distance, we give greater influence to these potentially under-represented domains, pulling the global model's convergence objective towards a more equitable and unbiased state. The drift-aware weight $w_k^{dist}$ is therefore computed as:

$$w_k^{dist} = \frac{d_k}{\sum_{j\in S_r} d_j}. \tag{2}$$

This mechanism ensures that domains which are further away from the global model are given a stronger voice in the aggregation, directly tackling the model aggregation bias and promoting a fairer learning process.

**Hybrid Weighting and Final Aggregation.** The final aggregation weight $w_k$ for each client is a linear combination of the conflict-aware and drift-aware weights, controlled by a hyperparameter $t \in [0, 1]$:

$$w_k = t \cdot w_k^{MOO} + (1 - t) \cdot w_k^{dist}. \tag{3}$$

The hyperparameter $t$ balances the two objectives: when $t \to 1$, the aggregation prioritizes minimizing objective conflicts based on parameter importance; when $t \to 0$, it prioritizes correcting for client drift. After computing the hybrid weights, they are re-normalized to sum to one. The server then performs the final aggregation to update the global model:

$$\boldsymbol{\theta}_{r+1} = \boldsymbol{\theta}_r - \sum_{k \in S_r} w_k \Delta_k. \tag{4}$$

The complete FedEquilibria procedure, which integrates client-side computation and the server-side hybrid aggregation, is summarized in Algorithm 1. By first computing conflict-aware weights via MOP and drift-aware weights via update norms, and then combining them through the hyperparameter $t$, our framework provides a principled and flexible mechanism for achieving fairness under domain skew.

---

**Algorithm 1** The FedEquilibria Algorithm

---

1: **Server Initializes:** Global model $\boldsymbol{\theta}_0$.
2: **for** each round $r = 0, 1, \ldots, T - 1$ **do**
3:     Server broadcasts $\boldsymbol{\theta}_r$ to a subset of clients $S_r$.
4:     **for** each client $k \in S_r$ **in parallel do**
5:         $(\Delta_k, \mathbf{F}_k) \leftarrow$ ClientUpdate$(k, \boldsymbol{\theta}_r)$.
6:     **end for**
7:     $\boldsymbol{w_k} \leftarrow$ Aggregation$(\boldsymbol{\theta}_r, \{(\Delta_k, \mathbf{F}_k)\})$.
8:     $\boldsymbol{\theta}_{r+1} \leftarrow \boldsymbol{\theta}_r - \sum_{k \in S_r} w_k \Delta_k$.
9: **end for**

10: **procedure** CLIENTUPDATE$(k, \boldsymbol{\theta}_r)$
11:     $\boldsymbol{\theta}_k^{r+1} \leftarrow$ Local training starting from $\boldsymbol{\theta}_r$.
12:     Compute model update $\Delta_k = \boldsymbol{\theta}_r - \boldsymbol{\theta}_k^{r+1}$
13:     Compute FIM profile $\mathbf{F}_k$
14:     **return** $(\Delta_k, \mathbf{F}_k)$.

15: **end procedure**

16: **procedure** AGGREGATION$(\boldsymbol{\theta}_r, \{(\Delta_k, \mathbf{F}_k)\})$
17:     $\{w_k^{MOO}\} \leftarrow$ SolveQP$(\{\mathbf{F}_k\})$
18:     $\{w_k^{dist}\} \leftarrow$ NormalizeNorms$(\{\Delta_k\})$
19:     $W_{total} \leftarrow 0$
20:     **for** each client $k \in S_r$ **do**
21:         $\tilde{w}_k \leftarrow t \cdot w_k^{MOO} + (1 - t) \cdot w_k^{dist}$
22:         $W_{total} \leftarrow W_{total} + \tilde{w}_k$
23:     **end for**
24:     **for** each client $k \in S_r$ **do**
25:         $w_k \leftarrow \tilde{w}_k / W_{total}$
26:     **end for**
27:     **return** $\boldsymbol{w_k}$.
28: **end procedure**

---

# 5 EXPERIMENT

## 5.1 EXPERIMENT SETTING

**Datasets and Models.** To simulate domain skew, we conduct experiments on three widely-used benchmark datasets. The first is Digits, a combination of four digit recognition datasets (MNIST, USPS, SVHN, and Synthetic Digits) where each serves as a client domain, for which we use a ResNet-10 (He et al. (2016)) model. The second is Office-Caltech (Gong et al. (2012)) , which contains images from 10 object categories across four domains (Amazon, Caltech, DSLR, and Webcam), evaluated with a ResNet-12 model . The third is PACS (Li et al. (2017)), a more challenging benchmark with four domains (Photo, Art Painting, Cartoon, and Sketch), also using a ResNet-12 model. We allocate 20 clients for each task and distribute an equal number of clients to each domain. We sample different proportions of the original training data to construct the client's local training set: 1% for Digits and 10% for both Office-Caltech and PACS.

**Implementation Details.** We compare our proposed method, FedEquilibria, against a comprehensive set of state-of-the-art federated learning algorithms. The baselines include the canonical

FedAvg (McMahan et al., 2017), FedProx (Li et al., 2020), MOON (Li et al., 2021a), FedProto (Tan et al., 2022), FedHEAL (Chen & Vikalo, 2024), FedLsa (Fu et al., 2025), and Pa3Fed (Huang & Liu, 2025). All experiments are implemented in PyTorch and run for a total of 200 communication rounds. For the Digits dataset, clients perform 10 local epochs of training in each round with a batch size of 64. For Office-Caltech and PACS, clients perform 5 local epochs with a batch size of 16. We use the SGD optimizer for all local training procedures, with a learning rate of 0.001, momentum of 0.9, and weight decay of 1e-5. For our method, FedEquilibria, the configuration of its hyperparameters is detailed in the Hyperparameter Study section.

**Evaluation Metrics.** To comprehensively evaluate performance and fairness, we use three key metrics calculated from the final global model's accuracy on each domain's test set. Average Accuracy (AVG) measures the mean accuracy across all domains to indicate overall performance. Standard Deviation (STD) of accuracies is used to quantify fairness; a lower value indicates smaller performance disparity and thus a fairer model. Finally, Worst-Client Accuracy (MIN), the minimum accuracy among all domains, directly measures the model's ability to not disadvantage any single client, a key goal in achieving fairness.

## 5.2 PERFORMANCE EVALUATION

Table 1: Main performance and fairness comparison on the Digits and Office-Caltech benchmarks. We report the final accuracy on each domain, along with the cross-domain average accuracy (AVG), standard deviation (STD), and worst-client accuracy (MIN). ↑ indicates that higher is better, while ↓ indicates lower is better. The best result in each summary column is highlighted in bold.

| Methods | Digits | | | | | | | Office-Caltech | | | | | | |
|---|---|---|---|---|---|---|---|---|---|---|---|---|---|---|
| | MNIST | USPS | SVHN | SYN | AVG↑ | STD↓ | MIN↑ | Amazon | DSLR | Caltech | Webcam | AVG↑ | STD↓ | MIN↑ |
| FedAvg | 94.44 | 93.17 | 79.07 | 39.60 | 76.57 | 22.18 | 39.60 | 65.79 | 66.67 | 54.91 | 43.10 | 57.62 | 9.57 | 43.10 |
| FedProx | 93.95 | 93.12 | 79.94 | 39.75 | 76.69 | 22.04 | 39.75 | 70.53 | 63.33 | 58.93 | 41.38 | 58.54 | 10.74 | 41.38 |
| MOON | 94.72 | 93.17 | 79.65 | 40.45 | 77.00 | 21.90 | 40.45 | 69.47 | 66.67 | 58.04 | 39.66 | 58.46 | 11.65 | 39.66 |
| FedProto | 94.26 | 92.53 | 80.51 | 38.70 | 76.50 | 22.46 | 38.70 | 71.05 | 66.67 | 60.71 | 44.83 | 60.82 | 9.93 | 44.83 |
| FedHEAL | 91.21 | 95.27 | 80.63 | 44.40 | 77.88 | 20.05 | 44.40 | 72.11 | 60.00 | 59.38 | 48.28 | 59.94 | 9.50 | 48.28 |
| Pa3Fed | 94.91 | 93.77 | 80.25 | 39.15 | 77.02 | 22.61 | 39.15 | 67.89 | 60.00 | 56.25 | 48.28 | 57.98 | **7.78** | 48.28 |
| FedLsa | 96.12 | 92.68 | 79.97 | 39.30 | 77.02 | 22.59 | 39.30 | 70.00 | 66.67 | 58.04 | 39.66 | 60.79 | 10.10 | 39.66 |
| FedEquilibria | 90.87 | 94.37 | 80.90 | 50.00 | **79.04** | **17.48** | **50.00** | 73.16 | 63.33 | 56.70 | 51.72 | **61.23** | 8.03 | **51.72** |

We evaluate the effectiveness of our proposed method, FedEquilibria, against a suite of state-of-the-art federated learning algorithms on three challenging domain skew benchmarks. The comprehensive results are presented in Table 1 and 2, where we assess performance based on average accuracy (AVG), standard deviation of accuracies (STD), and worst-client accuracy (MIN).

The results clearly demonstrate the superiority of FedEquilibria in achieving a better balance between overall performance and fairness. On the Digits dataset, FedEquilibria establishes a clear advantage. It achieves the highest average accuracy (79.04%) among all methods. More critically, it demonstrates a profound impact on fairness. The worst-client accuracy (MIN) is boosted to 50.00%, representing a significant improvement of over 5.6 percentage points compared to the next best method, FedHEAL (Chen & Vikalo (2024)). This improvement is directly attributable to our hybrid weighting mechanism, which prevents the model from overfitting to easier domains like MNIST and instead allocates more learning capacity to the challenging SYN domain. This fairness enhancement is further quantified by the standard deviation (STD), where FedEquilibria records the lowest value of 17.48, markedly reducing the performance disparity across domains. This strong performance extends to the more complex object recognition tasks. On Office-Caltech, FedEquilibria again leads in average accuracy (61.35%) and achieves the highest worst-client accuracy (50.00%). Similarly, on the most challenging PACS dataset, FedEquilibria obtains the highest average accuracy (59.82%) and dramatically lifts the worst-client performance to 47.34%, a gain of over 5 percentage points compared to the best baseline on this metric (Pa3Fed). While some methods like Pa3Fed occasionally achieve a slightly lower STD, they do so at the cost of significantly lower average and minimum accuracy, highlighting FedEquilibria's superior ability to improve fairness without sacrificing overall performance.

Table 2: Performance and fairness comparison on the PACS benchmark.

| Methods | PACS | | | | | | |
|---|---|---|---|---|---|---|---|
| | PHOTO | ART | CARTOON | SKETCH | AVG↑ | STD↓ | MIN↑ |
| FedAvg | 63.50 | 61.56 | 63.48 | 37.18 | 56.43 | 11.14 | 37.18 |
| FedProx | 66.77 | 67.15 | 64.12 | 33.38 | 57.58 | 14.18 | 33.38 |
| MOON | 68.84 | 60.83 | 64.12 | 33.25 | 56.76 | 13.87 | 33.25 |
| FedProto | 65.28 | 65.69 | 65.61 | 34.90 | 57.87 | 13.26 | 34.90 |
| FedHEAL | 56.68 | 59.12 | 70.70 | 40.23 | 56.68 | 10.88 | 40.23 |
| Pa3Fed | 58.16 | 60.34 | 65.82 | 42.13 | 56.61 | 8.81 | 42.13 |
| FedLsa | 63.50 | 60.10 | 64.33 | 38.58 | 56.63 | 10.54 | 38.58 |
| FedEquilibria | 67.95 | 58.39 | 65.61 | 47.34 | **59.82** | **8.02** | **47.34** |

The training dynamics on the Digits dataset, illustrated in Figure 2, provide further insight into our method's effectiveness. Figure 2(a) shows that FedEquilibria not only reaches the highest final accuracy but also maintains a competitive convergence trajectory throughout the training process. Concurrently, Figure 2(b) reveals the core strength of our framework: the standard deviation of accuracies for FedEquilibria is consistently and substantially lower than all other methods from the early stages of training. This demonstrates that our fairness-aware aggregation is not just a final-round phenomenon but an active, continuous process that effectively mitigates performance disparity throughout learning, leading to a robust and equitable final model.

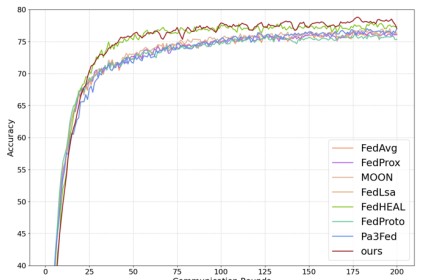
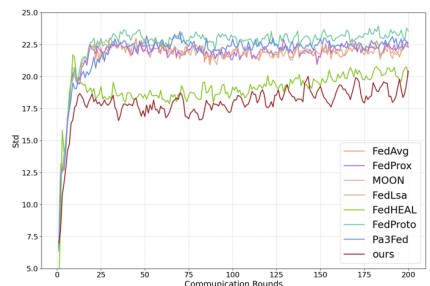

(a) Comparison of convergence of **average accuracy** with counterparts on Digits.    (b) Comparison of convergence of **standard deviation** with counterparts on Digits.

Figure 2: Comparison of convergence dynamics on the Digits dataset. (a) Average accuracy over 200 communication rounds. (b) Standard deviation of accuracies, illustrating the evolution of fairness.

## 5.3 ABLATION STUDY

Table 3: Ablation study of the key components of FedEquilibria on the Digits and Office-Caltech datasets. We evaluate the average accuracy (AVG) and standard deviation (STD). $MOO_G$ denotes using gradients for multi-objective optimization, $MOO_F$ denotes using the Fisher Information Matrix (FIM), and $DIST$ denotes the drift-aware distance weighting component. The first row (all components disabled) corresponds to the FedAvg baseline.

| $MOO_G$ | $MOO_F$ | $DIST$ | Digits | | | | | | Office-Caltech | | | | | |
|---|---|---|---|---|---|---|---|---|---|---|---|---|---|---|
| | | | MNIST | USPS | SVHN | SYN | AVG↑ | STD↓ | Amazon | DSLR | Caltech | Webcam | AVG↑ | STD↓ |
| | | | 94.44 | 93.17 | 79.07 | 39.60 | 76.57 | 22.18 | 65.79 | 66.67 | 54.91 | 43.10 | 57.62 | 9.57 |
| ✓ | | | 95.44 | 95.17 | 77.88 | 40.90 | 77.35 | 22.21 | 73.68 | 63.33 | 58.04 | 39.66 | 58.68 | 12.34 |
| | ✓ | | 90.14 | 95.17 | 78.09 | 48.00 | 77.85 | 18.32 | 65.79 | 70.00 | 54.91 | 46.55 | 59.31 | 9.20 |
| | | ✓ | 92.09 | 93.02 | 82.50 | 43.55 | 77.79 | 20.19 | 71.58 | 63.33 | 55.80 | 48.28 | 59.75 | 8.66 |
| | ✓ | ✓ | 90.87 | 94.37 | 80.90 | 50.00 | **79.04** | **17.48** | 73.16 | 63.33 | 56.70 | 51.72 | **61.23** | **8.03** |

To validate the effectiveness and necessity of each component in our proposed FedEquilibria framework, we conducted a comprehensive ablation study, with the results presented in Table 3. We analyze three core variants: using only the multi-objective optimization on gradients ($MOO_G$), using only MOO on the Fisher Information Matrix ($MOO_F$), and using only the distance-based weighting ($DIST$). The results clearly demonstrate that each component contributes positively, and their synergistic combination in FedEquilibria yields the best performance. Specifically, the

$DIST$ component alone improves upon the FedAvg baseline in both average accuracy and fairness (lower STD), confirming its effectiveness in mitigating model aggregation bias. More importantly, the results highlight the superiority of using the FIM for conflict resolution. The $MOO_F$ variant consistently outperforms the $MOO_G$ variant, achieving a significantly lower STD on both datasets (e.g., 18.32 vs. 22.21 on Digits). This provides strong empirical evidence for our central hypothesis: resolving conflicts in the parameter importance space (FIM) is more effective for achieving fairness than resolving conflicts in the gradient space. Finally, the full FedEquilibria model, which combines $MOO_F$ and $DIST$, achieves the highest average accuracy and the best fairness on both datasets, demonstrating that our two components are complementary and work synergistically to address the distinct challenges of update conflicts and client drift.

### 5.4 HYPERPARAMETER STUDY

The hyperparameter $t$ in Equation(3) is crucial as it balances the conflict-aware weights ($w_k^{MOO}$) and the drift-aware weights ($w_k^{dist}$). The former addresses parameter update conflicts by finding a fair consensus, while the latter counteracts model aggregation bias by accounting for domain diversity. We conduct an analysis to find the optimal value of $t$ for each dataset, with results shown in Figure 3. For the Digits and PACS datasets, the choice is straightforward. As shown in Figure 3(a) and 3(c), the optimal performance is achieved at $t = 0.7$ for Digits and $t = 0.8$ for PACS, respectively. At these values, both datasets achieve the highest average accuracy and the lowest standard deviation, indicating a clear optimal balance. The Office-Caltech dataset presents a more complex accuracy-fairness trade-off, a common challenge in heterogeneous settings. As seen in Figure 3(b), the peak average accuracy is achieved at $t = 0.3$, while the best fairness (lowest STD) occurs at $t = 0.7$. This divergence can be attributed to the high inter-domain variation within this benchmark. As $t$ increases, our model places greater emphasis on the multi-objective optimization component, which strongly enforces fairness by minimizing update conflicts, thus progressively driving down the STD. However, this intense focus on fairness can slightly pull the model away from the point of maximum average performance. Therefore, selecting an appropriate hyperparameter is vital for navigating this trade-off. For our main experiments, we select $t = 0.3$ for Office-Caltech, as it offers a compelling balance, retaining high accuracy while achieving a substantial improvement in fairness.

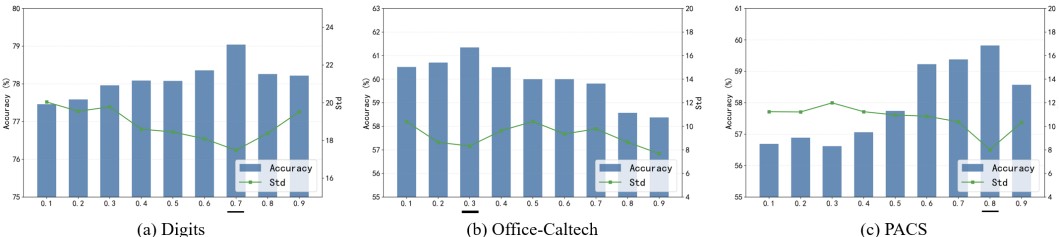

(a) Digits      (b) Office-Caltech      (c) PACS

Figure 3: Hyperparameter study on the effect of the balancing coefficient $t$ across the (a) Digits, (b) Office-Caltech, and (c) PACS datasets. The bars represent average accuracy (left y-axis), while the line plot shows the standard deviation of accuracies (right y-axis). The optimal values chosen for our main experiments are underlined.

## 6 CONCLUSION

In this work, we introduced FedEquilibria, a novel federated aggregation framework designed to address the critical challenge of performance fairness under domain skew. By identifying update conflicts and model aggregation bias as primary impediments, we proposed a synergistic hybrid weighting mechanism. Our approach uniquely models aggregation as a multi-objective optimization problem in the Fisher information space to find a consensus on parameter importance, while a complementary drift-aware component based on the update norm counteracts aggregation bias by amplifying the influence of under-represented domains. Extensive experiments demonstrate that FedEquilibria substantially outperforms state-of-the-art methods, achieving superior average accuracy while markedly improving fairness, most notably by significantly boosting worst-client performance. This work validates a principled approach to building robust and equitable global models in heterogeneous environments.

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

# A APPENDIX

## A.1 THE USE OF LARGE LANGUAGE MODELS

The language of this paper was polished using large language models (LLMs) to enhance clarity and readability. The final content and academic integrity remain the responsibility of the authors.

## A.2 DATASET DISTRIBUTION DETAILS

To provide a comprehensive overview of the data heterogeneity simulated in our experiments, this section details both the qualitative and quantitative aspects of our dataset setups.

We first illustrate the qualitative nature of the domain skew in Figure 4. The figure displays sample images from each domain within the three benchmarks. It is visually apparent that while the underlying classes are the same (e.g., digits, objects), the feature distributions—including style, background, color, and quality—vary dramatically from one domain to another. This visual disparity exemplifies the core challenge that federated learning algorithms face in this setting.

Beyond these qualitative differences, our experimental setup incorporates significant quantitative heterogeneity. Figure 5 provides a detailed visualization of the data distribution for each of the 20 clients across the three benchmarks. For each task, clients were partitioned by being assigned to a single domain (e.g., clients $C_{18}$ and $C_{19}$ are assigned to the MNIST and SYN domains, respectively, in the Digits task). The plots highlight two key aspects of the data skew: (1) a non-IID class distribution, as the number of samples per class (represented by colored segments) is unbalanced for each client, and (2) a data size imbalance, as the total number of samples (the total bar length) differs significantly between clients from different domains. This carefully constructed setup creates a challenging and realistic environment for rigorously evaluating the fairness and robustness of the compared algorithms.

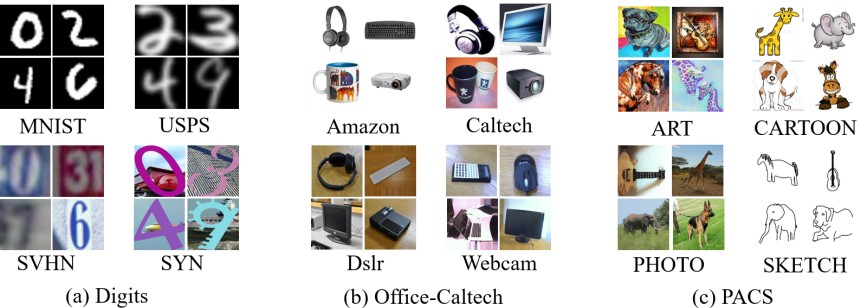

|            |          |            |            |            |            |
|------------|----------|------------|------------|------------|------------|
| MNIST      | USPS     | Amazon     | Caltech    | ART        | CARTOON    |
| SVHN       | SYN      | Dslr       | Webcam     | PHOTO      | SKETCH     |
| (a) Digits |          | (b) Office-Caltech |    | (c) PACS   |            |

Figure 4: Qualitative illustration of domain skew in the benchmark datasets. Each column showcases sample images from a distinct domain for (a) **Digits**, (b) **Office-Caltech**, and (c) **PACS**, highlighting the significant variations in visual features (e.g., style, color, perspective) despite sharing the same semantic classes.

## A.3 THE FISHER INFORMATION MATRIX IN DISTRIBUTED AND CONTINUAL LEARNING

The Fisher Information Matrix (FIM) has emerged as a powerful tool for quantifying parameter importance in neural networks, with its origins in continual learning. Elastic Weight Consolidation (EWC) Kirkpatrick et al. (2017) famously uses a diagonal approximation of the FIM to identify parameters crucial for previously learned tasks and penalizes changes to them to prevent catastrophic forgetting. This principle is broadly applicable, leveraging the FIM to obtain crucial information about the sensitivity of model parameters.

## A.4 TRAINING DYNAMICS ON THE OFFICE-CALTECH DATASET

To further investigate the performance of FedEquilibria on a more challenging benchmark, we present a detailed analysis of the training dynamics on the Office-Caltech dataset. Figure 6 plots the average accuracy and the standard deviation of accuracies over 200 communication rounds.

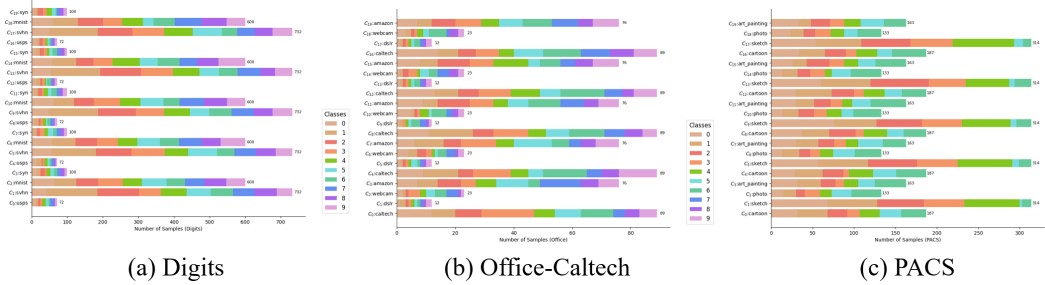

|  |  |  |
|---|---|---|
| (a) Digits | (b) Office-Caltech | (c) PACS |

Figure 5: Quantitative data distribution for each of the 20 clients across the (a) Digits, (b) Office-Caltech, and (c) PACS benchmarks. Each horizontal bar represents a single client, labeled with its index and assigned domain. The colored segments within each bar show the number of samples per class, illustrating the non-IID and unbalanced nature of the local datasets.

The results highlight the increased difficulty imposed by the significant domain skew in Office-Caltech, as evidenced by the highly volatile performance of most baseline methods. As shown in Figure 6(a), while many algorithms struggle with erratic fluctuations in accuracy, our method (ours) demonstrates a notably more stable convergence trajectory. After the initial training phase, FedEquilibria consistently maintains a position among the top-performing methods, avoiding the severe performance drops that affect other approaches.

The superiority of our framework is most evident in the evolution of fairness, as depicted in Figure 6(b). While all methods exhibit a sharp initial increase in standard deviation as client models begin to diverge, the baseline methods continue to show high and unstable STD values throughout the training process. In stark contrast, the curve for FedEquilibria is consistently and substantially lower and smoother than all competitors. This demonstrates that our hybrid aggregation mechanism is not only effective at achieving a fair outcome but also excels at maintaining it stably from round to round. This stability directly supports our central claim: by resolving conflicts in the more stable, structural space of parameter importance (FIM) rather than the noisy, instantaneous space of gradients, FedEquilibria achieves a more robust and principled path toward a fair and accurate global model.

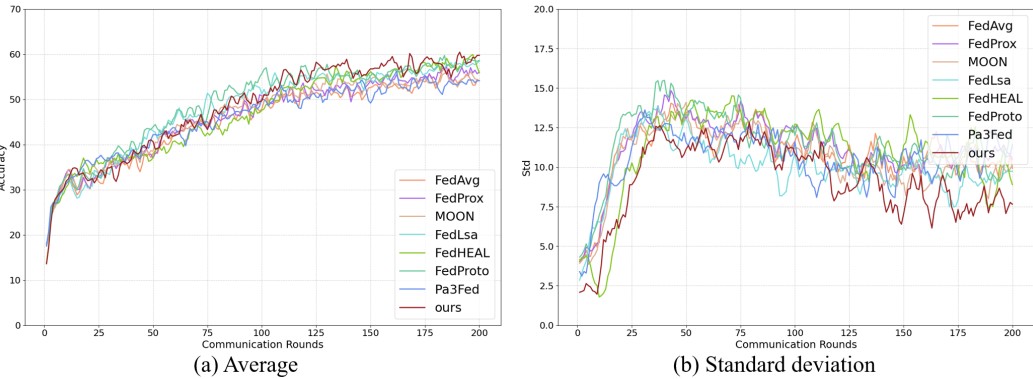

|  |  |
|---|---|
| (a) Average | (b) Standard deviation |

Figure 6: Comparison of training dynamics on the Office-Caltech dataset. (a) Average accuracy over 200 communication rounds. (b) Standard deviation of accuracies. The plots show that FedEquilibria (ours) achieves both a more stable convergence in accuracy and a consistently lower standard deviation compared to the volatile performance of the baseline methods.

## A.5 COMPLETE ABLATION STUDY RESULTS

To confirm that the conclusions from our main ablation study hold on more challenging data, we present the complete ablation results on the PACS dataset in Table 4. This benchmark is known for its significant inter-domain variance, making it an excellent testbed for evaluating the robustness of each component of our framework.

The results on PACS strongly reinforce our central claims. We observe that the drift-aware distance weighting component ($DIST$) alone provides a significant improvement over the FedAvg baseline, boosting average accuracy by over 2 percentage points and reducing the standard deviation from 11.14 to 10.31. This confirms its effectiveness in correcting for model aggregation bias. Crucially, the comparison between using FIM ($MOO_F$) and gradients ($MOO_G$) for multi-objective optimization highlights the superiority of our approach. While $MOO_F$ improves both average accuracy and fairness, the gradient-based $MOO_G$ actually worsens the standard deviation (12.39 vs. 11.14), despite a minor gain in average accuracy. This suggests that on complex datasets, simply reconciling noisy, instantaneous gradients can fail to improve fairness, whereas optimizing in the more stable, structural space of parameter importance provides a more robust path to an equitable solution. Finally, the full FedEquilibria model, which synergistically combines the $MOO_F$ and $DIST$ components, achieves the best results across the board, with the highest average accuracy (**59.82%**) and the lowest standard deviation (**8.02**). This demonstrates that the two components are not redundant but are complementary, addressing distinct facets of the fairness problem to produce the most effective and equitable global model.

Table 4: Ablation study of the key components of FedEquilibria on the PACS dataset. We evaluate the average accuracy (AVG) and standard deviation (STD). $MOO_G$ denotes using gradients for multi-objective optimization, $MOO_F$ denotes using the Fisher Information Matrix (FIM), and $DIST$ denotes the drift-aware distance weighting component. The first row (all components disabled) corresponds to the FedAvg baseline.

| $MOO_G$ | $MOO_F$ | $DIST$ | PHOTO | ART | CARTOON | SKETCH | AVG↑ | STD↓ |
|---|---|---|---|---|---|---|---|---|
| | | | 63.50 | 61.56 | 63.48 | 37.18 | 56.43 | 11.14 |
| ✓ | | | 56.97 | 64.23 | 71.13 | 37.94 | 57.57 | 12.39 |
| | ✓ | | 70.33 | 57.66 | 64.33 | 41.75 | 58.52 | 10.67 |
| | | ✓ | 69.14 | 57.18 | 65.61 | 42.39 | 58.58 | 10.31 |
| | ✓ | ✓ | 67.95 | 58.39 | 65.61 | 47.34 | **59.82** | **8.02** |

## A.6 FULL HYPERPARAMETER SETTINGS

For the sake of reproducibility, this section details the hyperparameter settings used for all baseline methods and our proposed framework, FedEquilibria. All common hyperparameters, unless otherwise specified, were shared across all methods. Specifically, all experiments were run for 200 communication rounds. The local training optimizer was SGD with a learning rate of 0.001, momentum of 0.9, and weight decay of 1e-5. The number of local epochs was set to 10 for the Digits dataset and 5 for both Office-Caltech and PACS. The batch size was 64 for Digits and 16 for Office-Caltech and PACS.

The method-specific hyperparameters, which were crucial for their performance, were tuned based on the recommendations from their respective original papers and our own validation experiments. The final values used for each benchmark are summarized in Table 5. For our method, FedEquilibria, the optimal values for the balancing coefficient $t$ were determined through the hyperparameter study detailed in the main paper.

Table 5: Method-specific hyperparameter settings for the baselines and our proposed method, Fed-Equilibria, across the three benchmark datasets. All other common settings (e.g., learning rate, local epochs) are described in the text.

| Method | Digits Setting | Office-Caltech Setting | PACS Setting |
|---|---|---|---|
| FedProx (Li et al., 2020) | $\mu = 1.0$ | $\mu = 1.0$ | $\mu = 1.0$ |
| MOON (Li et al., 2021a) | $\tau = 0.5, \mu = 5$ | $\tau = 0.5, \mu = 5$ | $\tau = 0.5, \mu = 5$ |
| FedProto (Tan et al., 2022) | $\lambda = 1.0$ | $\lambda = 1.0$ | $\lambda = 1.0$ |
| FedHEAL (Chen & Vikalo, 2024) | $\tau = 0.3, \beta = 0.4$ | $\tau = 0.4, \beta = 0.4$ | $\tau = 0.4, \beta = 0.4$ |
| Pa3Fed (Huang & Liu, 2025) | $\delta = 0.001, \beta = 0.3$ | $\delta = 0.01, \beta = 0.3$ | $\delta = 0.01, \beta = 0.3$ |
| FedLsa (Fu et al., 2025) | $\alpha = 0.02, \lambda = 0.1, \tau = 0.1$ | $\alpha = 0.02, \lambda = 0.7, \tau = 0.1$ | $\alpha = 0.02, \lambda = 0.1, \tau = 0.1$ |
| **FedEquilibria (Ours)** | $t = 0.7$ | $t = 0.3$ | $t = 0.8$ |

## A.7 COMPARISON WITH GRADIENT-BASED MULTI-OBJECTIVE OPTIMIZATION(FEDMGDA+)

To further validate our core design choice of using the Fisher Information Matrix (FIM) for multi-objective optimization, we conduct a direct comparison with FedMGDA+ (Hu et al., 2022a), a state-of-the-art FL algorithm that applies the MOO framework directly to client gradients. The primary goal of FedMGDA+ is to compute a common descent direction by solving a quadratic programming problem that minimizes the norm of the aggregated gradient vectors. This approach seeks a consensus on the update *direction*, ensuring that the global model update does not conflict with any client's local gradient. This makes it an ideal baseline for assessing the benefits of resolving conflicts in the parameter importance space (FIM) versus the gradient space.

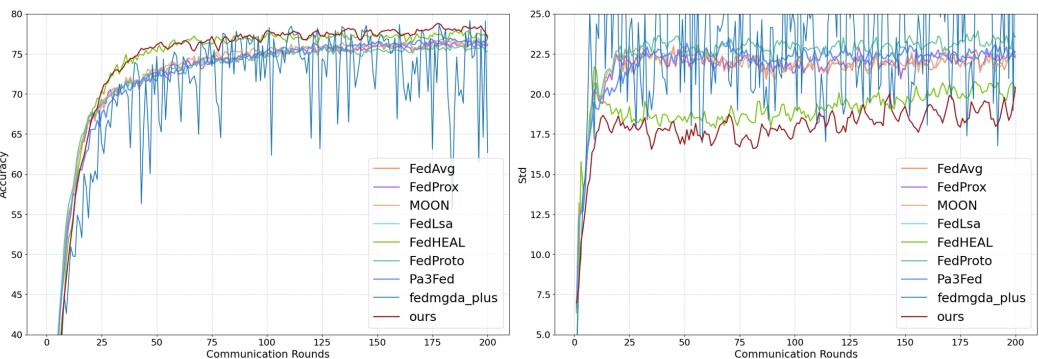

Figure 7: Training dynamics on the Digits dataset, including the gradient-based MOO method Fed-MGDA+. (Top) Average accuracy over 200 communication rounds. (Bottom) Standard deviation of accuracies. The plots highlight the significant instability inherent in gradient-based MOO compared to our FIM-based approach (ours).

The empirical results of this comparison, shown in Figure 7, reveal the significant limitations of relying on gradients alone. As seen in the top plot, the accuracy of FedMGDA+ is highly volatile throughout the training process, failing to converge to a stable and high-performing solution. This instability is even more pronounced in the context of fairness, as shown in the bottom plot. The standard deviation curve for FedMGDA+ is extremely erratic and consistently higher than that of our method, indicating that it struggles to maintain a fair performance distribution across clients from one round to the next. In contrast, our method, FedEquilibria, demonstrates smooth convergence for both accuracy and standard deviation, achieving a final state that is both more accurate and substantially fairer.

This observed instability can be attributed to the fundamental nature of gradients in heterogeneous settings. First, there is a key difference in stability and structure. Gradients reflect the model's instantaneous descent direction on a specific mini-batch of data and are thus highly susceptible to

stochastic noise. Consequently, the QP problem solved by FedMGDA+ is different and potentially volatile in every round. The FIM, however, is computed as an expectation over a client's entire local dataset, providing a stable and structural measure of parameter importance. Therefore, solving the MOO problem in the FIM space, as FedEquilibria does, leads to a more consistent and robust update direction. Second, our approach achieves a higher-level consensus. Gradient-based MOO seeks a consensus on update directions, which can be a low-level compromise. In contrast, FIM-based MOO seeks a consensus on parameter importance. This represents a more fundamental alignment, ensuring the global model update respects what each client's domain deems structurally critical, rather than just their immediate desired direction. This higher-level agreement is particularly crucial for achieving stable and fair convergence under pronounced domain skew.

## A.8 Per-Domain Performance Visualization

To provide a more granular view of how different methods handle performance fairness, we visualize the final test accuracy of the global model on each individual domain within our three benchmark datasets. Figure 8 plots the accuracy for FedAvg, FedHEAL, FedLsa, and our method, FedEquilibria. An ideal fairness-aware algorithm should not only achieve a high average performance but also exhibit a flatter line, indicating smaller performance gaps between the best and worst domains.

The results clearly illustrate the effectiveness of our approach in achieving performance equity. In the Digits benchmark (left), while all methods perform well on the easier MNIST and USPS domains, the baseline methods suffer a dramatic performance collapse on the challenging SYN domain. In contrast, FedEquilibria (ours) achieves a significantly higher accuracy on SYN, demonstrating its ability to prevent the model from neglecting the most difficult domain. This pattern is consistent across the Office-Caltech (center) and PACS (right) datasets. In both cases, the accuracy line for FedEquilibria is visibly flatter and often higher on the lowest-performing domains (e.g., Webcam and Sketch) compared to the baselines. This visualization provides direct, qualitative evidence that our hybrid weighting mechanism successfully lifts the performance floor for disadvantaged clients, leading to a more robust and equitable global model.

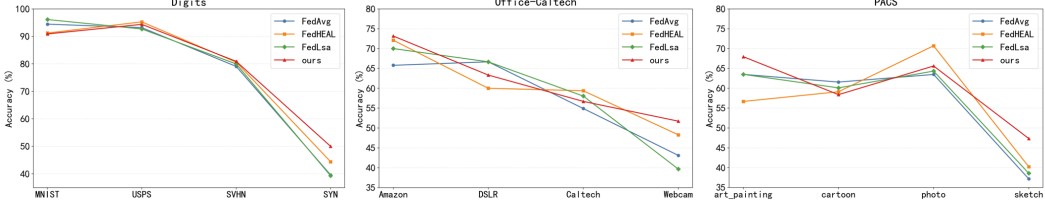

Figure 8: Per-domain final test accuracy on the (a) Digits, (b) Office-Caltech, and (c) PACS benchmarks. The x-axis lists the individual domains within each benchmark. A flatter line indicates better fairness (smaller performance disparity). Our method, FedEquilibria, consistently achieves higher accuracy on the most challenging domains (e.g., SYN, Webcam, Sketch), demonstrating superior performance fairness.

