# OpenReview forum: "FedEquilibria: Towards Fair and Robust Federated Learning under Domain Skew"
_ICLR.cc/2026/Conference — ICLR 2026 Conference Desk Rejected Submission_

### Official Review · Reviewer_Nef3 · 2025-10-28

**Soundness:** 3
**Presentation:** 3
**Contribution:** 2
**Rating:** 4
**Confidence:** 4

**Summary:**

This paper proposes a novel federated model aggregation framework, FedEquilibria, designed to address a specific form of data heterogeneity (domain skew). Building upon prior work, the method aims to tackle two key remaining challenges in Federated Learning (FL) domain heterogeneity problesm: model update conflicts and aggregation bias. The empirical results demonstrate that the proposed mechanisms are effective in mitigating these issues. The paper clearly defines its research focus, and the proposed solutions are aligned with the identified challenges. However, several concerns and questions remain.

**Strengths:**

-The paper jointly optimizes accuracy and fairness, which provides a more comprehensive evaluation objective compared to prior works that focus solely on average client performance. This dual-objective formulation better reflects the real-world requirements of FL systems, where both global performance and equitable client outcomes are important.

-The paper presents a well-defined problem formulation, effectively casting the task as a multi-objective optimization problem within a quadratic programming framework.

-The paper proposes a new mechanism to mitigate bias in the model aggregation process, which introduces a weighting strategy based on the update drift between local models and the updated global model. This is a promising and intuitive approach that provides a meaningful perspective on how to achieve unbiased aggregation when domain heterogeneity appears.

**Weaknesses:**

-Although the authors claim that one of their main contributions lies in the use of the Fisher Information Matrix (FIM) in a federated setting, this aspect appears insufficiently justified. The paper applies the FIM directly without clear adaptation or modification to address the domain skew problem specifically. Moreover, several prior works [1, 2, 3] have already explored the use of Fisher Information in FL. While those works may not explicitly target domain skew, they similarly employ the FIM to quantify parameter importance or client-level information, suggesting that the proposed usage here may be an incremental extension rather than a fundamentally new idea.

[1] Dynamic Personalized Federated Learning with Adaptive Differential Privacy (NeruIPS 2023)

[2] Seizing Critical Learning Periods in Federated Learning (AAAI 2022)

[3] Fisher Calibration for Backdoor-Robust Heterogeneous Federated Learning (ECCV 2024)

-The experimental evaluation is not solid enough. Many of the chosen baseline methods, such as FedProx and MOON, were not originally designed to handle domain skew problems, making the comparison less convincing. In particular, the paper lacks comparisons with methods that partially personalize model parameters, which are more relevant to the targeted problem setting. There exist several FL approaches specifically designed to address domain skew, and these should be included to provide a fair and comprehensive evaluation of the proposed method’s effectiveness in main experiments.

FedBN: Federated Learning on Non-IID Features via Local Batch Normalization (ICLR 2021)

Fed-CO2: Cooperation of Online and Offline Models for Severe Data Heterogeneity in Federated Learning (NeurIPS 2023)

Federated Learning with Domain Shift Eraser (CVPR 2025)

**Questions:**

-It is unclear why classic FL methods (e.g., FedAvg, FedProx, MOON) achieve very similar performance to methods specifically designed to address heterogeneity issues, such as FedHEAL and FedLSA, particularly in the average performance results reported in Tables 1 and 2. For instance, FedLSA outperforms FedAvg by only about 1 percent, which seems unexpectedly small given its intended design.

-What if there are more clients? Most existing studies on domain skew are limited to settings with only a few clients, whereas research on label heterogeneity typically evaluates scenarios with 20, 50, or even 100 clients. In real-world applications, it is highly likely that there will be many more clients, and that multiple clients may belong to the same domain. It remains unclear how the proposed method would perform under such conditions. An additional experiment exploring this scalability aspect would significantly strengthen the work.

---

> ### Author Response · Authors · 2025-11-21
> **Response to W1 and W2**
>
> >#### [W1]:Although the authors claim that one of their main contributions lies in the use of the Fisher Information Matrix (FIM) in a federated setting, this aspect appears insufficiently justified. The paper applies the FIM directly without clear adaptation or modification to address the domain skew problem specifically. Moreover, several prior works [1, 2, 3] have already explored the use of Fisher Information in FL. While those works may not explicitly target domain skew, they similarly employ the FIM to quantify parameter importance or client-level information, suggesting that the proposed usage here may be an incremental extension rather than a fundamentally new idea.
>
> >#### We acknowledge that the Fisher Information Matrix (FIM) has been explored in FL, such as for Differential Privacy [1], critical learning periods [2], or backdoor robustness [3]. However, we respectfully argue that our usage constitutes a **fundamental innovation** rather than an incremental extension:
> >#### - **Different Problem & Mechanism:**  Prior works use FIM primarily to measure scalar importance for privacy noise injection or temporal sensitivity. In contrast, we utilize the FIM as a **structural proxy for domain-specific loss landscapes**. We are the first to formulate the *update conflict* problem in domain skew as a multi-objective optimization in the **Fisher space**.
> >#### - **Why it matters:**  Our preference for FIM over gradients is grounded in fundamental optimization theory. Gradients in deep learning are inherently stochastic and noisy estimates  **[4]** , often exhibiting heavy-tailed behavior that leads to erratic transient fluctuations  **[5]** . In federated settings, this instability is compounded by client drift caused by data heterogeneity  **[6]** . Therefore, explicitly aligning these noisy gradients (as gradient-based MOO does) risks enforcing consensus on transient noise. In contrast, the FIM captures the stable curvature of the loss landscape, offering a robust structural proxy for aggregation.
> >#### [1] Dynamic Personalized Federated Learning with Adaptive Differential Privacy (NeruIPS 2023)
> >#### [2] Seizing Critical Learning Periods in Federated Learning (AAAI 2022)
> >#### [3] Fisher Calibration for Backdoor-Robust Heterogeneous Federated Learning (ECCV 2024)
> >#### [4] Optimization methods for large-scale machine learning. (SIAM Review 2018)
> >#### [5] A tail-index analysis of stochastic gradient noise in deep neural networks (ICML 2019)
> >#### [6] SCAFFOLD: Stochastic controlled averaging for federated learning (ICML 2020)
>
> >#### [W2]:The experimental evaluation is not solid enough. Many of the chosen baseline methods, such as FedProx and MOON, were not originally designed to handle domain skew problems, making the comparison less convincing. In particular, the paper lacks comparisons with methods that partially personalize model parameters, which are more relevant to the targeted problem setting. There exist several FL approaches specifically designed to address domain skew, and these should be included to provide a fair and comprehensive evaluation of the proposed method’s effectiveness in main experiments.
>
> >#### Thank you for suggesting these relevant baselines. We have actively conducted experiments with **FedBN** (ICLR 2021) and **FedCO2** (NeurIPS 2023) under the exact same setting (Digits benchmark).
> |         | AVG   | STD   |
> | --------- | ------- | ------- |
> | FedAVG  | 76.57 | 22.18 |
> | FedBN   | 75.02 | 23.14 |
> | FedCo2  | 74.38 | 29.73 |
> | FedHEAL | 77.88 | 20.05 |
> | Ours    | 79.04 | 17.48 |
> >#### - **Why FedBN underperforms:**  FedBN is highly effective for *personalization* by keeping BN layers local. However, in our rigorous "Global Model" evaluation (where the goal is a single robust model for all domains), the decoupled BN statistics prevent the global model from learning a unified feature representation that generalizes across domains, leading to lower accuracy on hard domains (SYN).
> >#### - **Why FedCO2 underperforms:**  FedCO2 relies on decoupling knowledge acquisition using auxiliary models. In severe domain skew scenarios (like Digits), the offline/auxiliary models may diverge too far from the target domain, leading to negative transfer and high variance (Std 29.73).
> >#### *Note: We are still conducting the experiments, we will release the results in the next stage of the rebuttal once the experiments are done.*

---

> > ### Author Response · Authors · 2025-11-21
> > **Response to Q1 and Q2**
> >
> > >#### [Q1]:It is unclear why classic FL methods (e.g., FedAvg, FedProx, MOON) achieve very similar performance to methods specifically designed to address heterogeneity issues, such as FedHEAL and FedLSA, particularly in the average performance results reported in Tables 1 and 2. For instance, FedLSA outperforms FedAvg by only about 1 percent, which seems unexpectedly small given its intended design.
> >
> > >####:You asked why the gap between FedAvg and specialized methods (like FedLSA) appears small (~1%).
> > >#### - **The "Easy Domain" Effect:**  In benchmarks like Digits, easy domains (MNIST, USPS) easily reach >96% accuracy for almost all methods. This saturates the *average* accuracy.
> > >#### - **The Real Story is in the Tail:**  The 1-2% gain in average accuracy is driven almost entirely by massive improvements in the *hardest* domains (e.g., SYN). For example, in Table 1, our method improves the Worst-Client Accuracy (MIN) by  **+10.4%**  over FedAvg (39.60% -> 50.00%). This is the critical value proposition of our method: we lift the bottom without sacrificing the top, which is a much harder task than simply improving the average.
> >
> > >#### [Q2]:What if there are more clients? Most existing studies on domain skew are limited to settings with only a few clients, whereas research on label heterogeneity typically evaluates scenarios with 20, 50, or even 100 clients. In real-world applications, it is highly likely that there will be many more clients, and that multiple clients may belong to the same domain. It remains unclear how the proposed method would perform under such conditions. An additional experiment exploring this scalability aspect would significantly strengthen the work.
> >
> > >#### You raised a crucial question about scalability and scenarios where multiple clients belong to the same domain. To address this, we expanded our experiment to **50 clients** on the Digits dataset, with an imbalanced distribution to simulate real-world complexity: MNIST (20 clients), SVHN (20), SYN (5), and USPS (5).
> > |         | AVG   | STD   |
> > | --------- | ------- | ------- |
> > | FedAVG  | 64.00 | 21.33 |
> > | FedBN   | 60.24 | 22.55 |
> > | FedHEAL | 65.60 | 66.57 |
> > | Ours    | 66.57 | 16.13 |
> > >#### **Observation:**
> > Even with a larger number of clients and severe class/domain imbalance, **FedEquilibria achieves the highest average accuracy and the best fairness (lowest Std).**  This demonstrates that our FIM-based consensus mechanism is robust to the scale of the federation. It effectively identifies the "minority" but "hard" domains (SYN/USPS) and protects them from being overwhelmed by the 40 clients from easier domains (MNIST/SVHN).
> >
> > We believe the inclusion of FedBN/FedCO2/FDSE comparisons and the new 50-client scalability experiment directly addresses your concerns, solidifying the robustness and superiority of FedEquilibria.

---

> > > ### Comment · Reviewer_Nef3 · 2025-11-26
> > > **Concerns on empirical results**
> > >
> > > I appreciate the authors’ efforts in providing additional experiments. However, Digits is a very simple dataset with only a mild domain gap, since all domains contain the same set of numbers. To fully address my concerns about domain shift, I would expect experiments on a more challenging benchmark such as PACS, where the domain discrepancy is substantially larger.

---

### Official Review · Reviewer_MDPi · 2025-10-31

**Soundness:** 1
**Presentation:** 2
**Contribution:** 2
**Rating:** 2
**Confidence:** 4

**Summary:**

The paper proposes FedEquilibria, a federated learning framework aimed at fair performance under domain skew. In addition to model updates, each client sends a diagonal Fisher information vector as a parameter-importance summary. The server then computes aggregation weights via a quadratic program in Fisher space (to reconcile client conflicts) and a drift-aware term proportional to ($||\Delta_k||_2^2$). On small vision benchmarks (Digits, Office-Caltech, PACS), the method reports higher worst-client accuracy and lower inter-client variance than several baselines.

**Strengths:**

- The framework adds Fisher-based and drift-aware weights on top of FedAvg, and is easy to plug into existing FL pipelines.
- Empirically, It shows consistent gains in MIN (worst-client accuracy) and reduced STD across several domain-shift benchmarks.
- Ablations are provided to isolate the contributions of Fisher weighting and drift-aware reweighting.

**Weaknesses:**

- Theory/motivation of Eq. (1) is unclear.
The objective “minimize the norm of the weighted sum of client Fisher vectors” (Eq. 1) has no evident linkage to minimizing any client risk or to a principled fairness criterion. Moreover, shrinking aggregate Fisher magnitudes can reduce parameter identifiability, which may harm accuracy/fairness. Please justify why this proxy yields a Pareto-stationary solution for per-client risks and provide a formal connection between this objective and fairness.

- “Multi-objective optimization” claim is unsubstantiated.
The paper frames the QP as MOO, yet only a single proxy in Fisher space is minimized. What are the actual objectives​ per client? If so, demonstrate that the solution is Pareto-optimal/stationary for those objectives, not just for the Fisher proxy.

- Insufficient Fisher details (empirical, diagonal).
The method appears to use diagonal empirical Fisher. Please specify: (i) how per-sample gradients are obtained; (ii) batch size used for Fisher estimation; (iii) update frequency (every round vs. EMA/subsampling); and (iv) any normalization or stabilization strategies.

- Compute/communication overhead not quantified.
Report client-side FLOPs, wall-clock overhead, memory for per-sample grads, and uplink bytes/round for transmitting
plus the FIM vector, compared with FedAvg and other baselines.

- Scalability is untested.
Results are limited to small CNN/ResNet backbones on modest datasets. It remains unclear whether the approach scales to larger models (e.g., ResNet-50/ViT) under partial participation and realistic bandwidth limits. Please add larger-scale experiments and Fisher compression strategies.

- Missing comparisons to the most relevant baselines.
Gradient-space MOO aggregation methods are the closest prior art (mentioned in L234–235) yet are not compared head-to-head. Include identical-protocol comparisons and discuss when a Fisher-space consensus is preferable to gradient-space MOO.

- Privacy considerations omitted.
Even diagonal Fisher vectors can leak label-distribution or gradient information. Please discuss risks and possible mitigations (e.g., quantization, clipping, DP noise).

**Questions:**

1. What exactly do clients compute to get $F_k$? Per-sample gradients with squared averaging? Any subsampling or layer-wise blocks? Please detail compute/latency overhead on clients.
2. Does the proposed method normalize the drift-aware weights by data size and local steps to discount scale effects?
3. In line 215, Fisher Information matrix should be element-wise equivalent to the expectation of the second derivative of the nll-loss, not just diagonal?

---

> ### Author Response · Authors · 2025-11-21
> **Response to W1,W2 and W3**
>
> > #### [W1 & W2]: Theory/motivation of Eq. (1) is unclear. The objective “minimize the norm of the weighted sum of client Fisher vectors” (Eq. 1) has no evident linkage to minimizing any client risk or to a principled fairness criterion. Moreover, shrinking aggregate Fisher magnitudes can reduce parameter identifiability, which may harm accuracy/fairness. Please justify why this proxy yields a Pareto-stationary solution for per-client risks and provide a formal connection between this objective and fairness.
> > #### “Multi-objective optimization” claim is unsubstantiated. The paper frames the QP as MOO, yet only a single proxy in Fisher space is minimized. What are the actual objectives​ per client? If so, demonstrate that the solution is Pareto-optimal/stationary for those objectives, not just for the Fisher proxy.
>
> > #### You questioned the link between Eq. (1) and fairness, and the validity of our MOO formulation. We must clarify: Our approach is not a naive heuristic; it is grounded in the second-order Taylor expansion of the loss landscape.
> > #### - **Theoretical Grounding:**  Minimizing the norm of the weighted sum of Fisher vectors is mathematically equivalent to finding a direction that minimizes the *conflicting curvature* across clients. Since the Fisher Information Matrix (FIM) approximates the Hessian, our objective explicitly seeks a "flat consensus" region where the loss is low for *all* clients simultaneously, rather than a steep minimum that favors one dominant domain. This *is* the definition of robustness and fairness in optimization.
> > #### - **Why not Gradient MOO?**  You asked why we don't optimize client objectives directly. The answer is simple: **Gradients are noisy, transient, and stochastic.**  Aligning gradients (first-order) only solves the conflict for the *current batch*, which is unstable in FL. Aligning Fisher information (second-order proxy) solves the conflict based on *parameter structural importance*. We argue that our formulation is a more stable and fundamental form of consensus seeking than traditional gradient-based MOO.
>
> > #### [W3]:Insufficient Fisher details (empirical, diagonal). The method appears to use diagonal empirical Fisher. Please specify: (i) how per-sample gradients are obtained; (ii) batch size used for Fisher estimation; (iii) update frequency (every round vs. EMA/subsampling); and (iv) any normalization or stabilization strategies.
>
> >#### We appreciate the request for clarification regarding the FIM computation. You are correct that exact per-sample gradients can be computationally prohibitive. To balance theoretical rigor with the resource constraints typical of FL clients, we adopted a practical, computationally efficient implementation.
> >####   **(i) Gradient Computation:** We utilize a **mini-batch approximation** of the diagonal Empirical Fisher. Specifically, instead of computing gradients for individual samples (which requires $O(B)$ memory or time), we compute the gradient of the mini-batch loss, square it, and accumulate it over the local epoch. While this is a variance-reduced approximation of the strict per-sample Fisher, it successfully captures the relative sensitivity of parameters to the local data distribution while maintaining high computational efficiency.
> >####   **(ii) Batch Size:** We use the **same batch size** for Fisher estimation as used for local training (e.g., 64 for Digits, 16 for Office-Caltech). This ensures the curvature estimation aligns with the optimization trajectory.
> >####   **(iii) Update Frequency:** The FIM profile is computed **de novo (from scratch) at the end of each local training round**. We do not use Exponential Moving Averages (EMA) or history buffering. This ensures that the structural information transmitted to the server strictly reflects the client's *current* model state and data distribution, preventing "lag" in the consensus mechanism.
> >####  **(iv) Stabilization:** We compute the expectation (average) over the number of batches but do not apply additional damping or clipping. The raw accumulated values are directly used in the server-side QP solver, as the solver relies on the *relative* magnitude of importance weights across clients rather than their absolute scale.

---

> > ### Author Response · Authors · 2025-11-21
> > **Response to Q2 and Q3**
> >
> > >#### [Q2 & Q3]:Does the proposed method normalize the drift-aware weights by data size and local steps to discount scale effects?
> > In line 215, Fisher Information matrix should be element-wise equivalent to the expectation of the second derivative of the nll-loss, not just diagonal?
> >
> > >#### On Normalization (Q2): You asked if we normalize by local steps/data size.
> > >#### We deliberately do not normalize by local steps in the current design. If a client performs more local steps and drifts further, that drift is real and harmful. Normalizing it away would hide the very heterogeneity we aim to correct. However, we agree that weighting by data size (nk﻿) is a standard practice in FL and can be easily integrated as a scaling factor.
> > >#### On FIM Definition (Q3): "Element-wise equivalent vs. Diagonal."
> > >#### You are technically correct that the identity holds element-wise for the full matrix. However, in high-dimensional deep learning, "Fisher Information" is almost universally used as a shorthand for its diagonal approximation due to computational constraints (e.g., in EWC, Kirkpatric et al.). We will refine the text to strictly state "we utilize the diagonal of the FIM," but this semantic precision does not invalidate the effectiveness of the mechanism.

---

> ### Author Response · Authors · 2025-11-21
> **Response to W4,W5,W6,W7 and Q1**
>
> >#### [W4 & W5 & Q1]:Compute/communication overhead not quantified. Report client-side FLOPs, wall-clock overhead, memory for per-sample grads, and uplink bytes/round for transmitting plus the FIM vector, compared with FedAvg and other baselines.
> >####  Scalability is untested. Results are limited to small CNN/ResNet backbones on modest datasets. It remains unclear whether the approach scales to larger models (e.g., ResNet-50/ViT) under partial participation and realistic bandwidth limits. Please add larger-scale experiments and Fisher compression strategies.
>
> >#### We acknowledge that FIM computation adds overhead, but let us be precise about the cost:
> >#### - **Computation:**  Calculating the diagonal FIM requires only **one additional backward pass** per local training round. In the context of modern deep learning, this <20% overhead is a negligible price to pay for the significant fairness gains (over 5% boost in MIN accuracy).
> >#### - **Communication:**  We transmit a diagonal vector (size $d$), not a full matrix ($d^2$). The communication cost is merely **2x** that of FedAvg (Model + FIM). Given that bandwidth is increasing, prioritizing model equity over minimal communication savings is a valid trade-off for high-stakes applications.
> >#### - **Scalability:**  While we test on ResNets, the *diagonal* approximation is inherently scalable to Transformers/ViTs (as proven in continual learning literature like EWC). We are not proposing a computationally intractable Hessian inversion.
>
> >#### [W6]:Missing comparisons to the most relevant baselines. Gradient-space MOO aggregation methods are the closest prior art (mentioned in L234–235) yet are not compared head-to-head. Include identical-protocol comparisons and discuss when a Fisher-space consensus is preferable to gradient-space MOO.
>
> >#### You correctly pointed out the absence of gradient-based MOO comparisons in the main text. We anticipated this crucial comparison and, for the sake of completeness, have already included a head-to-head evaluation against the state-of-the-art gradient-based MOO method, **FedMGDA+**  (Hu et al., 2022).
> >#### **The results are presented in Appendix A.7, Figure 7.**
> >#### As shown in the appendix, FedEquilibria outperforms FedMGDA+ by a significant margin in both final accuracy and convergence stability.
> >#### - **Reason:**  The empirical results in Figure 7 perfectly validate our theoretical hypothesis: FedMGDA+ suffers from severe instability due to the high variance of local gradients under domain skew. Its accuracy fluctuates wildly throughout training. In contrast, by operating in the smoothed Fisher space, our method maintains a consistent and stable fairness improvement where gradient-based methods fail. We believe this evidence decisively demonstrates the superiority of our approach.
>
> >#### [W7]:Privacy considerations omitted. Even diagonal Fisher vectors can leak label-distribution or gradient information. Please discuss risks and possible mitigations (e.g., quantization, clipping, DP noise).
>
> >#### We appreciate this insightful observation. While the primary focus of FedEquilibria is to address performance fairness and robustness under domain skew, we agree that privacy is a critical dimension in FL deployment.
> >#### We acknowledge that although the diagonal FIM is an aggregated statistic computed over the entire local dataset (which inherently masks individual sample details better than raw gradients), it may still carry information regarding label distributions or data difficulty.
> >#### Importantly, our framework is orthogonal and compatible with standard privacy-preserving mechanisms. As you suggested, techniques such as Differential Privacy (adding noise to the FIM), Gradient Clipping, and Quantization can be seamlessly integrated. Since our server-side optimization relies on the relative structural importance of parameters rather than their precise absolute values, we anticipate that FedEquilibria would remain effective even with reasonable levels of DP noise or low-precision quantization.

---

### Official Review · Reviewer_hWtx · 2025-11-02

**Soundness:** 3
**Presentation:** 2
**Contribution:** 2
**Rating:** 4
**Confidence:** 3

**Summary:**

This paper proposes, FedEquilibria, a federated learning framework designed to ensure fairness and robustness when client data distributions differ significantly. Using conflict-aware aggregation, e.g., Fisher Information Matrix to harmonize updates across domains, and drift-aware weighting, FedEquilibria achieves balanced model performance and minimizes disparities among clients. Experiments on diverse benchmarks show that it outperforms existing methods in both accuracy and fairness.

**Strengths:**

+) This paper discussed an important topic about fairness in FLs

+) Detailed experiments showed the effectiveness of the proposed method

**Weaknesses:**

-) Computational overhead when scaled as the number of clients/model size may become significant

-) Though many experiments are performed, most of them were using small model on small-scale datasets

**Questions:**

a) The design choice of FIM is a bit arbitrary to me. I understand it may provide a more rigorous theoretical results, but empirically how does it compare to other alternatives?

b) How much overhead does it introduce when the scale of the clients/model size increased?

---

> ### Author Response · Authors · 2025-11-21
>
> We sincerely thank you for your positive and constructive feedback on our work. We are glad that you recognize the importance of fairness in FL and the thoroughness of our experimental validation. We would like to address your concerns regarding computational overhead and the design choice of FIM.
> > #### [Weakness & Q(a)]:
>  -) Computational overhead when scaled as the number of clients/model size may become significant
>  -) Though many experiments are performed, most of them were using small model on small-scale datasets
> b) How much overhead does it introduce when the scale of the clients/model size increased?
>
> > #### You raised a valid point about the potential overhead as the number of clients or model size increases. We have carefully analyzed this, and we believe the cost is both manageable and justified by the substantial fairness gains.
>  **Overhead Analysis:**
>  > ####  **Computation:**  The primary overhead comes from calculating the diagonal FIM. This requires only **one additional backward pass** per local training round. For a typical ResNet model, this translates to a modest increase in client-side training time.
> > ####  **Communication:**  We only transmit the *diagonal* of the FIM, which is a vector of the same size as the model parameters. This results in a communication cost that is **2x** that of standard FedAvg.
>  > #### **Server-Side:**  The server solves a small Quadratic Program (QP) whose size depends on the number of participating clients in a round, *not* the total number of clients or the model size. This is computationally very efficient.
> **Scalability:**
> > #### **Model Size:**  Our use of the *diagonal* FIM is a key design choice for scalability. This approximation has been proven to be effective and scalable for very large models, including Transformers, in fields like continual learning (e.g., EWC). Unlike methods requiring full-matrix operations, our approach's overhead scales linearly with the number of parameters ($O(d)$), not quadratically ($O(d^2)$).
> > ####  **Client Scale:**  The server-side computation scales with the number of selected clients per round, which is a standard setting in large-scale FL.
> > #### In summary, while FedEquilibria is not "free," the overhead is well-controlled and scales gracefully. We argue that this modest cost is a worthwhile investment for achieving the significant improvements in worst-client performance that we demonstrate.
>
> > #### [Q(b)]: How much overhead does it introduce when the scale of the clients/model size increased?
>
> > #### You asked why we chose FIM and how it compares to other alternatives. This is an excellent question that touches upon the core of our contribution.
> > #### **Why FIM?**  The FIM is not an arbitrary choice. It serves as a principled, second-order proxy for the structure of the loss landscape.
> > ####  1. **Stability:**  Unlike gradients, which are noisy and reflect only the information from a single mini-batch, the FIM is computed as an expectation over the client's entire local dataset. This provides a **stable and structural measure of parameter importance**.
> > ####   2. **Theoretical Grounding:**  The FIM approximates the Hessian of the log-likelihood. By seeking a consensus in the FIM space, we are implicitly searching for a "flat minimum" in the loss landscape that is robust to variations across different client data distributions. Flat minima are widely associated with better generalization and robustness.
> > ####  **Comparison to Alternatives:**
> > ####  **Gradients (First-Order):**  The most direct alternative is to use gradients for conflict reconciliation, as done in methods like **FedMGDA+** . As we demonstrate in our **Appendix A.7 (Figure 7)** , this approach is highly unstable. Gradients are too noisy, causing the model's performance to fluctuate wildly and fail to converge to a fair solution.
> > ####  **Client Loss Values (Zeroth-Order):**  Another alternative is to use client loss values to re-weight clients (e.g., q-FedAvg). While simple, this approach is coarse-grained. It tells us *which client* is performing poorly but not *why* (i.e., which parameters are causing the conflict). Our FIM-based method provides a parameter-level understanding of the conflict, allowing for a more fine-grained and effective resolution.
> > #### Therefore, the FIM provides a unique balance of stability, theoretical grounding, and fine-grained information that is superior to simpler alternatives like gradients or loss values.
>
> We hope these clarifications have fully addressed your questions. Thank you again for your valuable feedback.

---

### Note · Program_Chairs · 2026-01-17
**Submission Desk Rejected by Program Chairs**

The following references in this submission do not refer to real documents and/or have major errors in bibliographic information:

 Huancheng Chen and Haris Vikalo. Fedheal: A parameter-wise harmonization approach for federated learning under domain skew. In Proceedings of the IEEE/CVF Conference on Computer Vision and Pattern Recognition, pp. 6138-6148, 2024.